# Prolonged standing reduces fasting plasma triglyceride but does not influence postprandial metabolism compared to prolonged sitting

**Charles K. Crawford**, **John D. Akins**, **Emre Vardarli**, **Anthony S. Wolfe**, **Edward F. Coyle** *

Human Performance Laboratory, Department of Kinesiology and Health Education, University of Texas at Austin, Austin, TX, United States of America

* coyle@austin.utexas.edu

**Data Availability Statement:** All relevant data are within the paper and its Supporting Information files.

## Abstract

Prolonged periods of sedentary behavior are linked to cardiometabolic disease independent of exercise and physical activity. This study examined the effects of posture by comparing one day of sitting (14.4 ± 0.3 h) to one day of standing (12.2 ± 0.1 h) on postprandial metabolism the following day. Eighteen subjects (9 men, 9 women; 24 ± 1 y) completed two trials (sit or stand) in a crossover design. The day after prolonged sitting or standing the subjects completed a postprandial high fat/glucose tolerance test, during which blood and expired gas was collected immediately before and hourly for 6 h after the ingestion of the test meal. Indirect calorimetry was used to measure substrate oxidation while plasma samples were analyzed for triglyceride, glucose, and insulin concentrations. Standing resulted in a lower fasting plasma triglyceride concentration (p = 0.021) which was primarily responsible for an 11.3% reduction in total area under the curve (p = 0.022) compared to sitting. However, no difference between trials in incremental area under the curve for plasma triglycerides was detected (p>0.05). There were no differences in substrate oxidation, plasma glucose concentration, or plasma insulin concentration (all p>0.05). These data demonstrate that 12 h of standing compared to 14 h of sitting has a small effect the next day by lowering fasting plasma triglyceride concentration, and this contributed to a 11.3% reduction in postprandial plasma triglyceride total area under the curve (p = 0.022) compared to sitting.

## Introduction

As the industrialized world makes technological advancements, people spend a majority of their time seated [1]. Epidemiological analysis points to prolonged sitting as a modifiable risk factor for cardiovascular disease (CVD), independent of many other common risk factors [2]. Therefore, interventions that reduce sitting time may prove beneficial to health [3]. The degree of postprandial lipemia (PPL), or the rise in plasma triglyceride concentration in the 6–8 h after eating, is also associated with CVD and atherosclerosis and is consequently used as a

**Funding:** The authors received no specific funding for this work.

**Competing interests:** The authors have declared that no competing interests exist.

surrogate marker of cardio-metabolic health [4]. It has been demonstrated that acute exercise induces a reduction in PPL [5, 6], [7], but recent evidence suggests that this effect can be attenuated by prolonged sitting [8, 9], [3].

Given that prolonged sitting increases CVD risk even in people who meet the recommended exercise guidelines [3, 10], it is practical to investigate the effects of reducing prolonged sitting. Standing is an alternative to sitting as it would not prohibit many of the activities performed while sitting. Additionally, it has been shown that breaking prolonged periods of sitting with standing can lead to reductions in postprandial plasma glucose concentration, but not triglyceride concentration in some populations[11, 12,13]. However, neither plasma glucose nor plasma triglyceride concentration are reduced in nonobese adults and obese [14]. Accordingly, the effects of standing on fasting and postprandial metabolism are not completely elucidated. Therefore, this study aimed to replace one 12-h day of prolonged sitting with standing rather than fragmenting these periods, with the idea if differences exist they may be small and require an excessively long duration in order to magnify any potential difference. The purpose of the study was to determine if standing diminishes the increases in plasma triglyceride, glucose, and insulin concentrations following a high fat/glucose tolerance test (HFGTT) administered the next day. We hypothesized that 12 h of prolonged standing would improve the postprandial metabolic responses compared to 14 h of prolonged sitting.

## Subjects and methods

### Subjects

Eighteen (9 males, 9 females) healthy, recreationally active but untrained individuals (age: 24 ± 1 years) completed two trials in a crossover experimental design. Each trial was separated by at least six days. BMI for subjects was 25.75 ± 1.14 kg/m$^2$. Subjects had no history of cardio-vascular or metabolic disease, or cigarette smoking. Participants were first notified of risks and procedures involved with the study. They then gave their written informed consent prior to participation. Additionally, each subject completed a health history and physical activity questionnaire before preliminary testing. This study was registered at clinicaltrials.gov under the identifier NCT03089437 and approved by the University of Texas at Austin Institutional Review Board. The study took place at the Human Performance Laboratory at the University of Texas at Austin.

### Experimental protocol

Each subject underwent preliminary metabolic testing prior to starting the experimental trials. Afterwards, each trial consisted of three phases: a controlled activity phase of two days, a standing or sitting intervention phase of 12 h on the third day, and a high fat/glucose tolerance test (HFGTT) phase lasting 6 h on the fourth day. Subjects were asked to refrain from exercise or consuming caffeine or alcohol from the start of the first controlled activity day to the end of the HFGTT (Fig 1). There was a minimum 6-day washout between trials.

### Preliminary testing

Each subject came to the laboratory for preliminary testing the morning after an 8-h fast. Subjects' weight and height were measured, then each subject sat for 30 minutes then stood for 30 minutes. Subjects were asked to relax and breathe normally while sitting or standing. Subjects rested for 20 minutes before gas samples were collected to allow for a steady state to be reached [15]. During the last ten minutes of each 30-minute period each subject breathed into a meteorological balloon using a one-way valve (Hans Rudolph, Kansas City, MO). Expired gas

**Fig 1. Experimental design.** Sit or Stand Intervention was performed in the laboratory for 12 h on Day 3. A High Fat/Glucose Tolerance Test (HFGTT) was performed on Day 4.

samples were analyzed with a mass spectrometer (Perkin-Elmer MGA 1100, St Louis, Missouri) for $O_2$, $CO_2$, and $N_2$ concentration and a spirometer (VacuMed, Ventura, CA) for volume. The data from these gas samples were used to determine caloric expenditure during sample collection using standard formulas for indirect calorimetry, which was extrapolated to estimate resting and standing metabolic rate [16]. This calculation of the resting metabolic rate was used to determine the caloric content of the meals provided during the intervention phase.

## Controlled activity phase

The controlled activity phase occurred during the first two days of each trial. Subjects arrived at the laboratory at approximately 08:00 h. They were given a pedometer (Omron, Kyoto, Japan) for visual feedback and asked to achieve 5,500–6,500 steps per day, which is concordant to a non-sedentary, low level of physical activity [17]. To track activity, each subject was also equipped with an activPALμ activity monitor (PAL Technologies Limited, Glasgow, UK) on the thigh. This monitor allowed for the tracking of posture and activity level using a combined inclinometer/accelerometer. This monitor was not removed during the entirety of the controlled activity phase. Subjects were asked to consume their regular diet, sans caffeine and alcohol, and log the time and contents of everything consumed into a provided food journal. Subjects were asked to repeat food consumption exactly during the second controlled activity phase.

## Sitting/Standing intervention phase

On the third day of each trial, subjects reported to the Human Performance Laboratory at 08:00 h following a minimum 8 h fast. They were then asked to stand on a 6 $ft^2$ cushioned mat or sit in a cushioned chair for 12 h total. During this time subjects read, spent time on a computer, or watched movies/television. Sitting and standing were interrupted only for visits to the toilet, but steps were minimized otherwise. When standing, the subjects were allowed to lean periodically on the desk holding their computer screen. Each subject was provided breakfast, lunch, a snack, and dinner. These foods contained a macronutrient content containing approximately 59% carbohydrate, 20% protein, and 21% fat, in line with a standard diet (US Department of Health and Human Services, 2011). The number of calories provided equaled each subject's estimated resting metabolic rate and was replicated for both trials. This induced near estimated energy balance in the sit trial, and slight negative energy balance in the stand trial (i.e.; 97.9 ± 25.1 kcal).

After the 12-h protocol, subjects were asked to continue wearing the activity monitor and to fast until returning to the laboratory at 08:00 h the next morning for an HFGTT. They were asked to rest when they returned home and to minimize standing in the time between the intervention and the HFGTT test the next morning.

### High fat tolerance test phase

At 08:00 h on the morning after the standing/sitting intervention phase, subjects arrived at the laboratory to begin the HFGTT phase for measurement of postprandial metabolism. Upon arrival, the activPAL activity monitor was removed followed by measurement of body weight. Subjects were then seated and an antecubital venous catheter for blood collection was inserted during each trial. After five minutes, a fasting blood sample was collected in a $K_2$ EDTA tube (BD Vacutainer, Franklin Lakes, New Jersey) which was promptly centrifuged at 3,000 rpm for 15 minutes at 4°C (Eppendorf, Hamburg, Germany). Plasma was then aliquoted into a micro-centrifuge tube, labeled, and stored at -80°C (Thermo Scientific, Waltham, Massachusetts) until later analysis.

Once subjects were seated for twenty minutes, a 10-minute expired gas sample was collected. Thereafter, subjects were given a high fat shake consisting of melted ice cream and heavy whipping cream (1.34 g/kg fat, 0.92, g/kg carbohydrate, 0.17 g/kg protein, and 15.8 kcal/kg). Subjects were asked to consume the shake within five minutes. After completion of the shake, 4 ml of blood were sampled hourly for 6 h. Ten-minute gas samples were also collected at 2, 4, and 6 h post-ingestion for determination of substrate oxidation via indirect calorimetry.

### Biochemical analysis

Plasma triglyceride and glucose concentrations were determined via spectrophotometry using commercially available kits (Pointe Scientific Inc., Canton, MI) while plasma insulin was measured with the use of a commercially available human insulin enzyme-linked immunosorbent assay (ELISA) kits (Rocky Mountain Diagnostics Inc., Colorado Springs, CO). All sample measurements were run in duplicate using a plate reader (Tecan Infinite 200 PRO, Tecan Group Ltd., Mannedorf, Switzerland).

### Statistical analysis

Elevations in postprandial plasma triglyceride, glucose, and insulin are presented as total area under the curve ($AUC_T$) and incremental area under the curve above the baseline value ($AUC_I$).

Paired samples t-tests were used to compare the total postprandial substrate oxidation of carbohydrate and fat, total standing time, total sitting time, daily step numbers, caloric intake, sitting and standing metabolic rate, as well as the fasted plasma concentrations, $AUC_T$, and $AUC_I$ of plasma triglycerides, glucose, and insulin. Postprandial substrate oxidation and concentrations of plasma triglyceride, glucose, and insulin during each HFGTT were analyzed using two-way ANOVA with repeated measures (trials and time). When interactions were significant, Bonferroni multiple comparisons analyses were conducted. AUCs were calculated and data were analyzed using Graphpad Prism 7 (Graphpad Software, San Diego, CA). Data are presented as means and standard error. The level of significance was set *a priori* at α = 0.05.

## Results

### Preliminary testing

When extrapolated over a full 24 h, the average seated metabolic rate was 2118 ± 101 kcal/day, and the estimated cost of the standing intervention was 2216 ± 118 kcal/day, which amounts to a 5% difference (p<0.01). Larger subjects had greater absolute differences in the energy cost of standing compared to sitting. There was a significant correlation between height and

postural metabolic difference (r = 0.784, p<0.001); the same was observed for weight
(r = 0.661, p<0.01).

## Energy intake

Caloric intake was not different during the sitting and standing intervention days (2131 ± 101
kcal/day). During the standing intervention subjects ate 86 ± 29 fewer kcals than their esti-
mated metabolic rate including 12 h of standing (2216 ± 118 kcal/day).

## Steps and posture distribution

There was no significant difference (p>0.05) in the number of steps that participants took on
control days between trials (Fig 2A). There was also no significant difference (p>0.05) in the
amount of time that participants spent standing during the control days for each trial (Fig 2B).

All measurements of sitting, standing, and stepping on the day of the sit/stand intervention
include the entirety of the day, not just the time spent in the laboratory. There was no signifi-
cant difference (p>0.05) in steps taken on the sitting or standing intervention days (Fig 2A).
As designed, subjects sat significantly more during the sitting trial (14.4 ± 0.3 h p<0.001) and
stood significantly more during the standing trial (12.2 ± 0.1 h p<0.001; Fig 2B).

## Postprandial metabolism

Immediately prior to the HFGTT, fasted values for absolute (Fig 3B) and relative fat oxidation
(percentage of energy derived from fat relative to total energy expenditure) (Fig 3A) were not
significantly different (p>0.05), nor was the fasted resting metabolic rate (p>0.05; Fig 3C).
During the HFGTT there was no significant effect of posture on relative (p>0.05; Fig 3A) or
absolute fat oxidation, (p>0.05; Fig 3B). There were also no significant effects of posture on
metabolic rate during the HFGTT, (p>0.05; Fig 3C).

## Plasma concentrations

Fasting plasma triglyceride concentration was significantly reduced by 13.8% (p = 0.021) in
standing compared to sitting at the start of the HFGTT (Fig 4A). Over the course of the
HFGTT, there was a main effect of posture on plasma triglyceride concentration (p = 0.033;
Fig 4A), with lower concentrations displayed with standing compared to sitting. After calculat-
ing the plasma triglyceride $AUC_T$, these results are further supported as the standing $AUC_T$
was 11.3% lower than the sitting $AUC_T$ (p = 0.022; Fig 5A). However, plasma triglyceride
$AUC_I$ was not different in standing vs. sitting (Fig 5D; p = 0.186). The lower fasting plasma tri-
glyceride concentration in the standing trial compared to the sitting trial (p = 0.021) appeared
largely responsible for the 11.3% reduction in total $AUC_T$ for plasma triglyceride.

For plasma glucose (Fig 5B) and insulin (Fig 5C), there were no significant differences
between sitting and standing in $AUC_T$ (p>0.05), nor $AUC_I$ between trials (p>0.05; Fig 5E and
5F). There were no significant treatment effects of posture (p>0.05; Fig 4B and 4C).

## Discussion

The public is being advised to reduce sitting time for health reasons and one possible alterna-
tive is to stand, which has given rise to the use of standing desks and other behavioral changes.
However, there is little evidence that standing is better than sitting in terms of improvements
in fat and carbohydrate metabolism [11, 12]. We recognize that standing for 12 straight hours
is impractical, but we reasoned that if a metabolic benefit of standing exists, the best first
approach should be extreme. In order to shed light on the topic, this study took an extreme

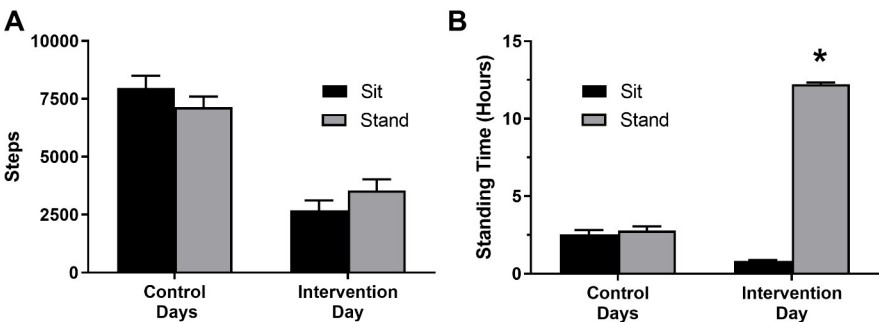

**Fig 2. Steps taken per day.** (**A**) and time spent standing (**B**) averaged over the two control days and the intervention day. (*): Significantly more time spent standing compared to the sitting trial (p<0.001).

approach by asking young healthy women and men to sit or stand for 12 h in the laboratory with postprandial metabolism measured the next day. More ecologically valid lengths of standing time have not shown improvements to postprandial triglyceride concentrations [14], but typically, a bout of even low intensity exercise compared to sitting, improves postprandial metabolism the next day in subjects who are physically active [7, 18]. In essence, our goal was to determine if prolonged static standing was metabolically different from sitting and if it elicited any similarities to previously described exercise responses.

The primary finding of this study is that a 12 h day of prolonged standing, compared to prolonged sitting, had relatively little influence on postprandial metabolism measured the following day. The postprandial increases in plasma glucose and insulin were very similar as was the oxidation of fat and carbohydrate. The primary effect of standing compared to sitting was that it lowered fasting plasma triglyceride concentration the following morning and this attenuation remained generally evident throughout the 6 h postprandial period resulting in a total plasma triglyceride $AUC_T$ that was lowered by 11.3% lower (p = 0.022) compared to sitting. However, the $AUC_I$ for plasma triglyceride was not different between trials, and fat oxidation was similar between trials during fasted and postprandial periods. This is further support for the idea that prolonged standing does not meaningfully improve postprandial substrate metabolism compared to prolonged sitting.

Acute and chronic periods of prolonged sedentary activity have been shown to induce a negative effect on postprandial metabolism. Early reports noted that one month of bed rest increases plasma triglyceride concentration in the fasted and the postprandial state [19,20]. Further evidence suggested that postprandial triglyceride concentration can increase after just two weeks of sedentary activity, even though subjects lost lean and total body mass, thus indicating an independence from positive energy balance [21]. More recent observations have found that only two days of prolonged sitting (1,700 steps/day) markedly worsened the postprandial metabolic response compared to an active (17,000 steps/day) condition; the two days of prolonged sitting were enough to prevent any exercise-induced decrease in postprandial lipemia despite one hour of moderate intensity exercise and a caloric deficit [8]. These findings came despite the multitude of studies that link exercise bouts to improved postprandial lipid profiles [5–7, 22, 23]. Even just one single day of prolonged sitting can negatively influence plasma triglyceride concentration [24, 25].

With sitting and standing there is a lack of locomotive muscle activity, though standing elicits postural muscular activity [26]. Our observation that energy expenditure when standing was only approximately 5% higher (p<0.01) than sitting is in concordance with previous reports of an 8% and 12% higher energy expenditure with standing [27, 28]. These present data highlight that standing has a small effect on metabolism the next day, but it does lower

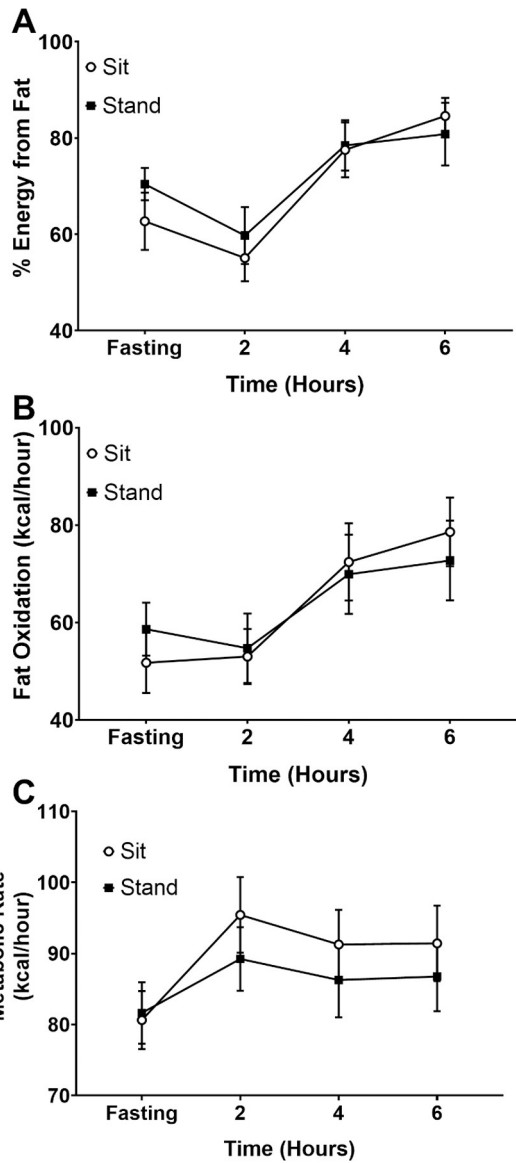

**Fig 3. Metabolism during the HFGTT.** Relative (**A**) and absolute (**B**) fat oxidation as well as total metabolic rate (**C**) during the HFGTT. Relative fat oxidation is the percentage of energy expenditure from fat. No significant differences were observed.

fasting plasma triglyceride concentration by 13.8% (p = 0.021) and $AUC_T$ by 11.3% (p = 0.022) in a young and healthy population with clinically normal fasting plasma triglyceride concentration [29]. Furthermore, subjects expended only 5% more calories, or fewer than 100 kcal, when standing compared to sitting for 12-h. It is therefore unlikely that the observed changes were caused by energy balance, given that the lowest energy expenditure difference needed to observe changes in postprandial lipemia has been reported to be approximately 200–250 kcal [30, 31]. With little movement and energy expenditure during either standing or sitting, this study demonstrates only a small effect of standing on raising metabolism compared to loco-motive muscular activity and the well-studied exercise-induced benefits [7]. The present study observed an 11.3% reduction in $AUC_T$ and no significant decrease in $AUC_I$. While this study did not directly compare the benefits of standing to light physical activity or exercise,

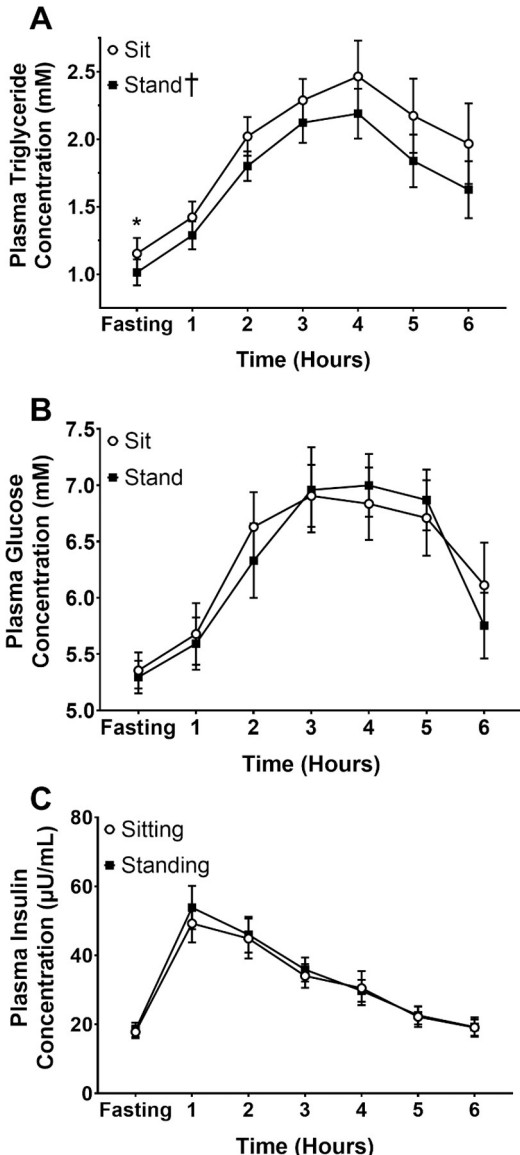

**Fig 4. Plasma concentrations during the HFGTT.** Plasma triglyceride (**A**), glucose (**B**), and insulin (**C**) concentration over the course of the High Fat/Glucose Tolerance Test. (†): Significant main effect of Standing compared to Sitting on lowering plasma triglyceride concentration (p = 0.033). (*): Significantly lower fasting plasma triglyceride concentration compared to the sitting trial (p = 0.021).

speculative comparisons may be drawn from other studies. Light physical activity is associated with a lower risk of exhibiting elevated plasma triglyceride concentrations [32,33]. High intensity exercise has been shown to decrease $AUC_T$ by 30.6%, and high, moderate, and low intensity exercise have been shown to decrease $AUC_I$ by 45.1%, 33.6%, and 17.2%, respectively [34, 18].

The 12 h of standing did yield a significant decrease in the $AUC_T$ during the HFGTT, but no significant decrease in $AUC_I$. $AUC_I$ is evaluated because it better describes the postprandial clearance of an oral fat load from the blood, while $AUC_T$ is descriptive of the total lipid profile including the fasting triglyceride value and not just the elevation from eating [35]. In other words, $AUC_T$ is influenced by the absolute concentration and fasted values, while $AUC_I$ is

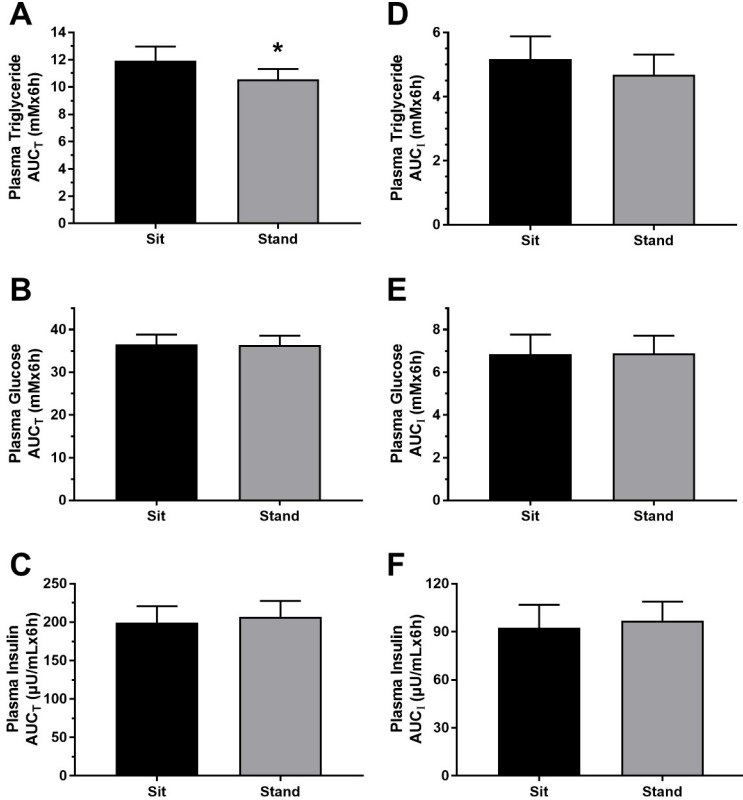

**Fig 5. Area under the curve for HFGTT.** Total Area Under the Curve ($AUC_T$) for plasma triglycerides (**A**), glucose (**B**), and insulin (**C**); Incremental Area Under the Curve ($AUC_I$) for plasma triglycerides (**D**), glucose (**E**), and insulin (**F**) during the HFGTT. (*): Significantly lower $AUC_T$ for plasma triglyceride concentration in the standing compared to the sitting trial (p = 0.022).

dependent on the rise in concentration [36]. Therefore, it appears that prolonged standing attenuates the total plasma triglyceride profile but does not significantly influence the incremental magnitude of the increase after a meal, specifically. Given that epidemiological studies have shown an association between the absolute values for both fasting and postprandial plasma triglyceride concentration and increased risk of CVD [37], it is possible that the lowering of $AUC_T$ for plasma triglycerides with prolonged standing may have health benefits even in the absence of a significant improvement in $AUC_I$ for plasma triglycerides.

By nature of the study, it is difficult to decipher the mechanisms responsible for the reduction in fasting plasma triglycerides or $AUC_T$ observed with standing compared to sitting. One speculation involves stored intramyocellular triglycerides, which are stored within muscle fibers and have a rate of uptake influenced by muscular contraction [38]. It is possible that postural contraction during prolonged standing decreased intramyocellular triglyceride content in postural muscles relative to the sitting intervention. Plasma triglyceride uptake into these muscles may have increased prior to the HFGTT, thus decreasing fasting plasma triglyceride concentration without increasing triglyceride oxidation. Future studies may utilize muscle samples to further understand the role of intramyocellular triglycerides in prolonged or intermittent standing compared to sitting.

Although not directly measured in this study, it is possible that prolonged standing alters the activity of lipoprotein lipase (LPL), which is the rate limiting enzyme for the plasma clearance of chylomicrons and very low density lipoprotein (VLDL) triglycerides [39]. Additionally,

LPL activity is shown to decrease following sedentary behavior but be highest with low intensity ambulatory activity and the activation of postural muscles [40, 41]. It is possible that the inactivation of postural muscles during prolonged sitting could reduce muscular LPL activity and impair the clearance of triglyceride from the blood into muscle. If the standing period did induce an increase in LPL, it could have induced a measurable lowering effect on fasting circulating triglycerides.

Insulin did not appear to be an influencing factor as plasma insulin concentrations were not different between trials in this study.

The present study is not without limitations. First, LPL was not directly measured; therefore, it is difficult to know if it accounted for the lower fasting plasma triglyceride concentration with prolonged standing compared to sitting. Second, in six subjects, hyperventilation was noted resulting in a high respiratory exchange ratio, and their data for calculating fat oxidation were eliminated. Third, there was not a control for the menstrual cycle in female subjects, though research suggests that menstrual phase would not alter lipid metabolism [42]. Fourth, this study only focused on one of the detrimental outcomes to prolonged sedentary behavior. Prolonged sitting is associated with problems including endothelial dysfunction, sleeping heart rate variability, and lower back pain, but these factors were not considered in this analysis [43, 44, 45, 46]. Finally, the subjects of this study were young healthy adults, and it is possible that these findings do not apply younger, older, or diseased populations.

In conclusion, these data show that 12 h of prolonged standing compared to 14 h of sitting significantly attenuates (11.3%, $p = 0.022$) the postprandial plasma triglyceride $AUC_T$ measured the next day without showing significant improvements in postprandial plasma triglyceride $AUC_I$. Fasting plasma triglycerides were reduced after standing, and this lower baseline appears to be the driving factor in the reduction of $AUC_T$. There were no differences between standing and sitting in postprandial fat oxidation, plasma glucose or plasma insulin responses. These findings indicate that for young adults with clinically normal fasting plasma triglyceride concentrations, standing compared to sitting for 12–14 h has a small benefit by lowering fasting plasma triglyceride concentration the next day. However, it does not appear to have appreciable benefits to postprandial metabolism such as those seen with exercise, especially of high intensity. For some, standing may be a way to introduce more active behaviors, but due to the arduous nature of standing for 12 h and the greater efficacy of physical activity and exercise compared to standing for lowering plasma triglyceride, physical activity or exercise appears to be a preferable intervention when attempting to counteract the unhealthy metabolic effects of prolonged sitting.

## Supporting information

**S1 Table. This is IRB approved protocol.** This shows the protocol as approved by the University of Texas at Austin's Institutional Review Board.
(PDF)

**S2 Table. This is TIDieR-checklist-word.** This checklist provides locations of information regarding elements of experimental protocol.
(DOCX)

**S3 Table. This is metabolic data GP.** This is a GraphPad file of metabolic data collected during the study.
(PZFX)

**S4 Table. This is activPAL data.** This is a GraphPad file of all data collected by activPAL devices during the study.
(PZFX)

## Acknowledgments

We thank the research subjects for their participation in the study.

## Author Contributions

**Conceptualization:** Charles K. Crawford, Edward F. Coyle.

**Data curation:** Charles K. Crawford, John D. Akins, Emre Vardarli.

**Formal analysis:** Charles K. Crawford, Anthony S. Wolfe.

**Funding acquisition:** Charles K. Crawford.

**Investigation:** Charles K. Crawford, John D. Akins, Emre Vardarli.

**Methodology:** Charles K. Crawford, John D. Akins, Emre Vardarli, Anthony S. Wolfe, Edward F. Coyle.

**Project administration:** Charles K. Crawford, John D. Akins, Emre Vardarli, Anthony S. Wolfe, Edward F. Coyle.

**Resources:** Charles K. Crawford, Edward F. Coyle.

**Software:** Charles K. Crawford.

**Supervision:** Charles K. Crawford, Edward F. Coyle.

**Validation:** Charles K. Crawford, John D. Akins.

**Visualization:** Charles K. Crawford, Edward F. Coyle.

**Writing – original draft:** Charles K. Crawford.

**Writing – review & editing:** Charles K. Crawford, John D. Akins, Emre Vardarli, Anthony S. Wolfe, Edward F. Coyle.

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
