## [Decision Letter · Decision Letter 0]

30 Oct 2019

PONE-D-19-20277

Prolonged standing reduces fasting plasma triglyceride but does not influence postprandial metabolism compared to prolonged sitting

PLOS ONE

Dear Dr. Coyle,

Thank you for submitting your manuscript to PLOS ONE. After careful consideration, we feel that it has merit but does not fully meet PLOS ONE’s publication criteria as it currently stands. Therefore, we invite you to submit a revised version of the manuscript that addresses the points raised during the review process.

We would appreciate receiving your revised manuscript by Dec 14 2019 11:59PM. To enhance the reproducibility of your results, we recommend that if applicable you deposit your laboratory protocols in protocols.io, where a protocol can be assigned its own identifier (DOI) such that it can be cited independently in the future. For instructions see: http://journals.plos.org/plosone/s/submission-guidelines#loc-laboratory-protocols

We look forward to receiving your revised manuscript.

Kind regards,

Martin Senechal, PhD

Academic Editor

PLOS ONE

Journal Requirements:

Reviewers' comments:

Reviewer's Responses to Questions

**Comments to the Author**

1. Is the manuscript technically sound, and do the data support the conclusions?

Reviewer #1: Yes

Reviewer #2: Yes

Reviewer #3: Yes

2. Has the statistical analysis been performed appropriately and rigorously? 

Reviewer #1: Yes

Reviewer #2: Yes

Reviewer #3: Yes

3. Have the authors made all data underlying the findings in their manuscript fully available?

Reviewer #1: Yes

Reviewer #2: Yes

Reviewer #3: Yes

4. Is the manuscript presented in an intelligible fashion and written in standard English?

Reviewer #1: Yes

Reviewer #2: Yes

Reviewer #3: Yes

5. Review Comments to the Author

Reviewer #1: The authors detail the findings from well-designed randomized cross-over trial that sought to examine the impact of prolonged standing versus prolonged sitting on energy in normal weight, healthy younger adults. The investigation of an extreme period (~12 hrs) of standing (as opposed to sitting), is a point of difference from previous investigations that have examined intermittent periods of shorter-duration standing in a fixed position. The manuscript is very well written and the authors are to be commended for the rigorous scientific methods applied.

The rationale for the study is well articulated, covering the current evidence that exists on this topic and what evidence gap this study sought to address.

Some aspects of the manuscript that warrant further attention include:

1. Methods: Recreationally active, but untrained participants were recruited – what screening processes were undertaken to ascertain current activity status?

2. Methods: It would be helpful to have additional description on what tasks the participants performed during the sitting/standing conditions? Were they able to work on a computer? How was this standardized between the two conditions?

3. Results: A significant difference was observed for the AUCT but not the AUCI for triglycerides. For the latter, the p value was 0.186. Is it possible that this reflects a statistical power issue – ie: if the sample size was larger, could it be possible that significant differences may have been observed?

4. Discussion: The authors have not addressed some of the key limitations of the study, specifically the restricted generalisabilty of the study since only healthy, generally normal weight younger adults were examined. This leads one to query whether similar findings are likely to be different in middle/older-aged adults and those who are overweight?

5. Discussion: Greater acknowledgement of the other potential detrimental consequences of extreme periods of prolonged standing (and also extreme periods of sitting for that matter) beyond that of metabolism needs to be given. For example the deleterious consequences on vascular physiology need to be acknowledged.

6. Discussion: The authors provide comparisons between the small changes observed in this trial with changes that have been observed with high intensity exercise. However, in the absence of study designs that have provided a direct head-to-head comparison, the comparability of the findings remains speculative.

7. Discussion: as acknowledged by the authors, an extreme bout of standing (and sitting) was used in this study. It would be helpful to have some further narrative on the extent to which this approach is ecologically valid and to speculate on what might occur with shorter (more ecologically valid) periods of sitting/standing.

Reviewer #2: This study examined the effects of posture by comparing one day of sitting to one day of standing on postprandial metabolism the following day. Despite a low ecological validity, because it is not feasible to ask people to be standing 12 hours a day, the verified results are interesting and clinically relevant, adding value for future studies of sedentary behaviour, with a reinforcing effect on the necessity of more active lifestyle as a tool for a good health.

Line 69: Why not the same period of time for both body position? Lines 117 – 118 are stated 12 hours for each behavior. Please, justify

Line 74- The term “recreationally active” means that the whole sample complied with physical activity recommendation for adults? Please make a clear definition on sample physical activity level!

Line 90 – Refrain for exercíse was the only recommendation? What about caffeine and alcohol ingestion?

Line 98 – Metabolic rates procedures are not adequately described!

Please describe a timeline and sequence of events before the indirect calorimetry procedure. This is really importante because the outcomes can be affected if the sample does not comply with all recommendations.

The Steady state was considered in gas Exchange analysis?

Which coeficiente of variation (CV) was achieved?

Line 109 – What criteria was used for a valid wear time?

How many hours a day ActivePal was used?

How many days of use?

The inclinometer was used to identified sitting time in this phase?

What physical activity level was identified in this sample?

All of them achieved physical activity recommendations or not?

Line 217 – Change fig 4D to fig 5D

Line 280 - Despite a small effect of standing on metabolism in the next day, this was enough to lower fasting glucose triglyceride concentrations by 13,8%. It is important to consider that the sample were comprised by young, health, normal-weight adults. Please reframe, talk about clinical relevance.

Line 289 – Will be interesting to discuss here about LIPA benefits as well. It is not a surprise that small muscle contractions in postural activities (standing) will generate smaller metabolismo alterations when compared with any exercise intensity. Following two suggested papers.

Amagasa S, Machida M, Fukushima N, Kikuchi H, Takamiya T, Odagiri Y, et al. Is objectively measured light-intensity physical activity associated with health outcomes after adjustment for moderate-to-vigorous physical activity in adults? A systematic review. Int J Behav Nutr Phys Act. 2018;15(1):65. doi: 10.1186/s12966-018-0695-z.

Howard B, Winkler EA, Sethi P, Carson V, Ridgers ND, Salmon JO, et al. Associations of Low- and High-Intensity Light Activity with Cardiometabolic Biomarkers. Med Sci Sports Exerc. 2015;47(10):2093-101. doi: 10.1249/MSS.0000000000000631.

Line 333 – it is important to highlight how many subjects were excluded due hyperventilation at methods section.

Line 345 – make clear if this small benefit has clinical relevance at this point too.

Line 347 – it is relevant to highlight here that standing it is not a more relevant strategy to blunts the increases in postprandial lipemia, but can be used to introduce changes, as an open window to achieve more active behaviors.

Reviewer #3: Review of Manuscript for PlosOne

Ref.: PONE-D-19-20277

Thank you for the opportunity to review your paper entitled: ‘Prolonged standing reduces fasting plasma triglyceride but does not influence postprandial metabolism compared to prolonged sitting’. The manuscript provides detail of a small study assessing the impact of prolonged standing on postprandial metabolic responses compared to long periods of sitting. This paper communicates original research, is reasonably well written, however there are several parts of the paper, which require some revision. I hope you find comments below helpful.

Abstract

The abstract is well written, I do not have any further suggestions for revision.

Introduction

Edits

Line 46- change end of sentence to: as a modifiable risk factor…

Line 56-57 suggest removing the sentence which assumes standing is the easiest alternative to sitting- this sentence is not referenced and there could be exceptions to this sentence (e.g. people with back pain may prefer moving to standing).

Lines 57 to 60 have used references from studies with diverse populations (e.g. pos-menopausal groups) and some are quite old. Can I suggest you read a recent systematic review by Saunders – Acute SB and markers of Cardiometabolic mix- it may assist you to find more appropriate references.

Line 63- while from an experimental point of view I can see why you would completely replace 14 hours of sitting with 12 hours of standing, I am not quite sure how this type of research would translate into the real world. There could be a number of occupational health and safety risks associated with prolonged standing, potentially breaking the standing with short periods of sitting may be more realistic. There is also some research which indicates excessive standing may impact cardiometabolic research Smith et al The Relationship Between Occupational Standing and Sitting and Incident Heart Disease. Suggest acknowledging this and then justifying your research in this regard.

Line 67 the word ‘blunts’ needs to be revised.

Subjects & Methods

Were the subjects required to have normal BMI?

Line 75 suggest you don’t need all of the information re: mean of height and weight as this is not relevant- BMI is more relevant.

Lines 77-78 needs rewording

Line 79- suggest indicating if the study risks and procedures were communicated prior to them consenting to participate.

Experimental protocol

Subjects undertook preliminary testing- but is doesn’t say what they were tested for.

Line 88 – replace sitting/standing with sitting or standing

The remainder of this section is well written.

Results

Suggest adding titles to your Figures to make them easier to interpret. With the information relating to the figures below them.

Discussion

The majority of the discussion is well written.

As per earlier suggestion- suggest revising the first sentence- re: natural alternative is to stand, especially given that the natural alternative would not be to stand for 12 hours straight. I acknowledge that you have recognised standing is not practical in the discussion- suggest stating this earlier in the manuscript.

Lines 247-250 is a good justification of your study – suggest stating this earlier.

Line 296- suggest replacing the word feeding with eating

Line – 305- suggest rewording this sentence (particularly the word illuminate)

Lines 307-314- suggest adding to this section how this might affect a future study- e.g. re: assessing the effects of frequent bouts compared to constant standing

Note: the discussion is quite long- lines 315-328 (as LPL was not measured in the study), suggest condensing this information into a couple of sentences.

Line 333- how many subjects experienced hyperventilation?

Lines 345- 350- suggest more thought in this section- physical activity and exercise- describe the types of activity (LPL) or potential for breaking up sitting with bouts that would be more appropriate in the context of the workplace or where sitting is likely to occur.

Line 351 onwards is repetitious suggest removing it.

6. PLOS authors have the option to publish the peer review history of their article (what does this mean?). If published, this will include your full peer review and any attached files.

Reviewer #1: No

Reviewer #2: No

Reviewer #3: Yes: Anne-Maree Parrish

---

## [Author Response · Author response to Decision Letter 0]

9 Jan 2020

All comments from reviewers have been addressed in the 'Response to Reviewers' letter attached in the submission.

---

## [Editor Report · Decision Letter 1]

13 Jan 2020

Prolonged standing reduces fasting plasma triglyceride but does not influence postprandial metabolism compared to prolonged sitting

PONE-D-19-20277R1

Dear Dr. Coyle,

We are pleased to inform you that your manuscript has been judged scientifically suitable for publication and will be formally accepted for publication once it complies with all outstanding technical requirements.

With kind regards,

Martin Senechal, PhD

Academic Editor

PLOS ONE
---

## [Editor Report · Acceptance letter]

29 Jan 2020

PONE-D-19-20277R1 

Prolonged standing reduces fasting plasma triglyceride but does not influence postprandial metabolism compared to prolonged sitting 

Dear Dr. Coyle:

I am pleased to inform you that your manuscript has been deemed suitable for publication in PLOS ONE. Congratulations! Your manuscript is now with our production department. 

With kind regards,

on behalf of

Dr. Martin Senechal 

Academic Editor

PLOS ONE